# Validation of the Generalized Workplace Harassment Questionnaire for Use with Medical Students

**DOI:** 10.3390/bs13100791

**Published:** 2023-09-23

**Authors:** Marcus A. Henning, Christian U. Krägeloh, Yan Chen, Fiona Moir, Craig S. Webster

**Affiliations:** 1Centre for Medical and Health Sciences, Faculty of Medical and Health Sciences, University of Auckland, Auckland 1010, New Zealand; yan.chen@auckland.ac.nz (Y.C.);; 2Department of Psychology and Neuroscience, Auckland University of Technology, Auckland 1010, New Zealand; chris.krageloh@aut.ac.nz; 3Department of General Practice and Primary Healthcare, Faculty of Medical and Health Sciences, University of Auckland, Auckland 1010, New Zealand

**Keywords:** medical students, Generalized Workplace Harassment Questionnaire, validation study

## Abstract

The Generalized Workplace Harassment Questionnaire (GWHQ) has not been validated among medical students whilst they are on clinical placements. Therefore, this study aims to validate its use when applied to this cohort. A sample of 205 medical students in their clinical training phase completed the GWHQ. To examine the validity of the proposed factor structure of the validated 25-item GWHQ, which was reduced to from the original 29-item set, a confirmatory factor analysis was conducted. Model fit was appraised by evaluating the comparative fit index (CFI), the root mean square error of approximation (RMSEA), and the standardized root mean squared residual (SRMR). Spearman’s rho correlation coefficients were applied to correlations between factors. With the exclusion of Item 19, the resulting fit was improved. In the revised model for a 24-item GWHQ, CFI = 0.995, RMSEA = 0.047, and SRMR = 0.115. Overall, the fit met the criteria for two fit indices and was thus deemed to be acceptable. Factor loadings ranged from 0.49 to 0.96. The Spearman’s rho correlation coefficient between Verbal and Covert Hostility was high, although all correlations with Physical Hostility were weak. In conclusion, the amended 24-item version of the GWHQ is a valid instrument for appraising instances of harassment or hostility within clinical placements attended by medical students in New Zealand.

## 1. Introduction

Medical students must participate in clinical placements as experiential workplace learning given this is a requirement of the degree. There has been considerable discourse pertaining to the workplace climate of clinical settings and how students often encounter negative experiences that impair their learning process [1,2]. Colenbrander, Causer, and Haire [1], in their qualitative study, outlined the consequences of medical students either witnessing or directly experiencing harassment during their clinical placements, culminating in detrimental impacts to their physical and mental wellbeing. Henning and colleagues [2] conducted a scoping review regarding medical students’ experiences of harassment during clinical placement, finding that sources of harassment stemmed from inherent and persistent toxic learning environments that reinforced abuse and misuse of power. These toxic workplace environments tended to be places that fostered sexual, physical, and verbal harassment, culminating in physical and mental health issues and problems associated with reinforcing unprofessional practices. More specifically, Lin, Rospenda, and Richman [3] reported that need for social approval and the experience of harassment were associated with alcohol misuse in male college students, rather than their female peers, whereas female students are more often at risk of experiencing sexual harassment than their male counterparts. The experience of sexual harassment often has a severe consequential effect in that it creates more internalized psychological distress which can lead to severe consequences, such as suicidal ideation and suicide attempts. Therefore, Lin and colleagues surmised that men are more likely to use alcohol to cope with harassment experiences whilst female students are more likely to cope through seeking help and rumination. These studies evidence that harassment or hostility is a problem existing in clinical and university settings, which likely has a harmful impact on students’ wellbeing. 

Auditing workplace practices requires systems to measure behavioral dynamics in the workplace so that appropriate interventions and policies can be implemented to minimize the incidence of harassment. It is crucial that measurements are developed and psychometrically evaluated so that interventions can be rigorously appraised. This will provide managers with the capacity to establish whether an intervention is effective or not. In addition, a psychometrically robust measurement will enable managers to confidently assess whether healthy practices are being maintained in their workplace environments [2]. In a recent systematic review [4], various measurements of workplace violence in healthcare settings were documented, including questionnaires measuring aspects of psychological distress, overt aggression, and mobbing. Other commonly cited measures used in higher education settings focus on aspects of sexual harassment, generalized workplace abuse, workplace incivility, and workplace bullying [5]. 

In this study, we focused on assessing the value of the Generalized Workplace Harassment Questionnaire (GWHQ) given its usage in numerous organizational settings [6], such as higher education. The GWHQ has been clearly established as an instrument of choice in numerous recent research projects. Four examples of its application include investigating the employment experiences of college students in the US [7], the workplace harassment experiences of university employees in the US [8], the workplace experiences of full-time working women in Pakistan and the mediating effect of cognitive hardiness [9], and family communication and wellness and links with corporal punishment, child abuse, and bullying in Ukraine [10]. Therefore, in recent years, there has been clear evidence of the GWHQ’s global utility within the educational and other organizational sectors.

However, this instrument has not been evaluated in reference to its utility in measuring harassment experiences reported by medical students during their training in clinical workplace settings. We envisaged that validating a measurement instrument to investigate workplace behaviors in clinical settings, such as harassment, will be valuable for organizational management in healthcare training venues, training and development in health-related educational facilities, and comparative research purposes in medical education. Therefore, this study aimed to validate the GWHQ to determine its utility in assessing harassment behaviors experienced by medical students during clinical placements.

## 2. Methods

### 2.1. Ethics Statement

The study was approved by the University of Auckland Human Participants Ethics Committee (Reference Number 023525), whereby informed consent and anonymity were assured.

### 2.2. Study Design

This is a cross-sectional validation study aimed to confirm the utility of the GWHQ for the medical student population in New Zealand during clinical training.

### 2.3. Participants

All students (N = 810) in their clinical training (4th to 6th years) of the Bachelor of Medicine and Bachelor of Surgery (MBChB) program at a university in New Zealand (2019) were approached via email and invited to participate in the survey.

### 2.4. Setting

The survey was conducted after final exams for that year given the time constraints faced by students during this phase of their training.

### 2.5. Measure

The GWHQ was the instrument under investigation in this study [6]. This questionnaire was originally comprised of 29 items related to aspects of workplace harassment and hostility. However, Rospenda and Richman reduced the original 29 items to 25 items after completing a domain-item validation study [6]. The GWHQ is purported to measure generalized workplace harassment [8], which “is comprised of four factors: covert hostility (e.g., excluded from important meetings or events, 3 items), verbal hostility (e.g., yelled at, talked down to, 7 items), manipulation (attempts to control the target’s behavior, e.g., through threats or bribes, 5 items), and physical aggression (e.g., pushed, hit, kicked, 1 item)”. In previous research, the reduced 25 items demonstrated four clearly and psychometrically defined factors, namely Verbal Hostility, Covert Hostility, Manipulation, and Physical Hostility [6]. The GWHQ clearly demonstrates that generalized workplace harassment is a multifaceted construct aligned with the four aforementioned factors. In this study, we applied a five-point Likert scale (Never = 1 to 5 = Very often), with a higher score indicating a greater occurrence of hostility or harassment behavior.

### 2.6. Statistical Methods

A confirmatory factor analysis (CFA) was conducted using LISREL 8.80 software [11]. The estimation method of diagonally weighted least squares with polychoric correlations was applied, which is considered to be the most appropriate approach for ordinal-level datasets such as those from Likert-scale items [12,13]. Our investigative approach was specifically selected as the most appropriate one for this analysis as the scale was presented on a 5-point Likert scale, as mentioned above. Additionally, it was also expected that several items would involve specific actions unlikely to have occurred frequently (e.g., Item 25: “…threw something at you?”), thus resulting in floor effects. The use of polychoric correlations is recommended when data do not meet the assumptions of normality and may even be used for Likert response scales with as few as 2 or 3 options [12,13].

The magnitude of the discrepancy between the sample and the fitted covariance matrix is often evaluated using chi-square, where a good model provides a nonsignificant (*p* > 0.05) result. However, because of the tendency of chi-square to become inflated with large sample sizes [14,15,16], it is common practice to evaluate goodness of fit using other indices instead, including the comparative fit index (CFI), the root mean square error of approximation (RMSEA), and the standardized root mean squared residual (SRMR). In the present study, model fit was considered acceptable if CFI > 0.96, RMSEA < 0.06, and SRMR < 0.09, which reflects commonly accepted cut-off criteria [17,18]. Our analysis tested a four-factor model where the factors Verbal Hostility, Covert Hostility, Manipulation, and Physical Hostility were correlated. Modification indices were inspected throughout the iterative analysis process. However, even if modification indices indicated that the correlation of item error co-variances will result in an improved fit, such correlations were only considered for correlations within factors. Even though the CFA cut-off values are relevant at two decimal places, we reported these values to three decimal places so that the proximity of results to these cut-off values can be illustrated more accurately.

## 3. Results

### 3.1. Participant Details

A total of 205 medical students completed the online survey (response rate = 25%). The average age of respondents was 23.75 years (SD = 2.91). More female students (68.8%) than male students responded to the survey, which is consistent with the total population of medical students in these cohorts (i.e., 55%). There were representative responses in reference to years of study (Year 4 = 83, Year 5 = 62, Year 6 = 60) and across ethnic groupings (Māori = 20, Pacific Islands = 13, Asian = 70, New Zealand European = 90, Other = 12). Statistical subgroup inferential analyses were not conducted given that this study aimed to validate the questionnaire with respect to the medical student sample as a whole. We saw further subgroup analyses to be an area for investigation once the questionnaire had been validated.

### 3.2. Confirmatory Factor Analysis

Running the initial four-factor model resulted in an acceptable fit according to the CFI (1.00) and RMSEA (0.052; 90% CI 0.043; 0.062), although the SRMR was elevated with a value of 0.127. Factor loadings ranged from 0.58 to 0.96 and were thus acceptable. The exception was Item 19 (i.e., “…offered you a subtle or obvious bribe to do something that you did not agree with?”), which had a value of 0.34. As a result of this low factor loading, the next iteration of the psychometric analysis was conducted without Item 19.

With the exclusion of Item 19, the resulting fit with correlated error variances was slightly improved; CFI continued to be very high at 0.995, RMSEA decreased to 0.047 (90% CI 0.036; 0.057), and SRMR decreased slightly to 0.115. Although the SRMR was still above the recommended cut-off of 0.080, the fit met the criteria for the other two fit indices and was thus deemed to be acceptable overall. Factor loading values ranged from 0.49 to 0.96, with the majority of items above 0.70 (Table 1). Spearman’s rho correlation coefficients for the subscale scores are shown in Table 2. The correlation between Verbal Hostility and Covert Hostility was very high, and all correlations with Physical Hostility were weak.

## 4. Discussion

The primary purpose of this study was to conduct a psychometric evaluation of the GWHQ using confirmatory factor analysis techniques, after which the questionnaire could be used with more confidence in further medical education research or be employed by managers within healthcare settings. A detailed scoping review [2] identified the antecedents (e.g., being exposed to toxic workplace cultural practices) and the consequences (e.g., depleted physical and psychological health) of harassment with specific reference to the medical students’ clinical experience. This scoping review provided a conceptual backdrop for this study and was the motivation for the current empirical research.

Measuring and appraising levels of harassment within workplace settings are acknowledged as crucial processes to ensure a safe and productive organizational milieu [19]. Assessing the occurrence of harassment incidents can also alert managers to the presence of toxic elements in their workplace, which likely has a deleterious impact on the wellbeing of key stakeholders [2]. In the clinical learning environment, these stakeholders can be identified as health professionals, administrators, patients, service personnel, and visiting students. As a consequence, this study aimed to focus on the experiences of medical students within clinical settings.

In reference to the medical student sample, the response rate may be considered contentious; however, the sample size was sufficient to enable a CFA to be conducted [20]. Overall, the results indicated that the GWHQ demonstrated a good model fit and thus was able to reasonably measure a generalized facet of harassment (or hostility) within clinical settings for the medical students in this study. As shown in Table 1, the four-domain structure was confirmed as well formed and the coefficient values for the factor loadings were adjudged to be acceptable and comparable to other study results using CFA [21]. More specifically, we applied the four-factor model with correlated factors but did not compute a higher-order factor. Most factor loadings were above 0.70. The exception was Item 19 (bribes), which had a loading of 0.34. When we deleted this item, the fit was improved. We surmised that the word *bribe* may be perceived as a very specific and underutilized term in the New Zealand vernacular, and hence this item obtained a strong floor effect. We evidenced that this is the most likely reason explaining why the *bribe* item did not fit well (i.e., 199 chose 1, 4 chose 2, and only 2 people chose 3; nobody chose 4 or 5 on this item).

The findings of this study indicate that the 24-item GWHQ, reduced from the original validated 25-items, will likely be a valuable and reliable instrument that can accurately appraise medical students’ experiences of harassment during their clinical training and be used in future research to assess different subgroups that may be considered at more risk of harassment, such as female medical students [8]. This information will be informative for clinical leaders, curriculum planners, clinical teachers, healthcare managers, student advocates, and medical students. The information can be supplemental to other more specific questionnaires, such as employing a sexual harassment questionnaire [22]. Data obtained from the GWHQ can be combined with other measures to create a more detailed picture of organizational workplace dynamics, thus identifying whether marginalized groups are more at risk of harassment, and whether harassment can have consequential outcomes such as toxic team-based behaviors, power imbalance, psychological and physical health issues, lowered performance, and safety concerns [2]. This research could incorporate the GWHQ to develop more detailed deterministic models using structural equation modeling. Evaluating the prevalence of harassment behaviors within the clinical setting is, thus, clearly informative as the first step in ensuring that medical students are learning in safe, work-oriented educational settings. Identification of toxic learning environments needs to be aligned with subsequent and immediate action to ameliorate hostility issues or prevent student allocation to unsafe clinical workplaces.

The fit of the CFA was excellent according to two of the three goodness-of-fit indices, although the SRMR remained elevated. For a scale of that nature, where floor effects and thus severe deviations from normality are expected, the statistical approach that we used is considered to be the most appropriate [13]. In this context, the results appear robust but will nevertheless need to be replicated with other samples, particularly with larger sample sizes to ensure a wider spread of scores despite the floor effects that are inherent in this scale. The results of the factor analysis also indicate that some items will likely not work with medical student cohorts in the New Zealand clinical context, such as Item 19. This may be due to cultural and language idiosyncrasies pertinent to different regions and professions. Nonetheless, since the study was conducted with medical students in New Zealand, the findings indicate that it will likely be fit for purpose for similar cohorts in the Australasian region. The moderately low response rate (25%) may be perceived as a limitation, although good representation was noted across gender, years of study, and ethnicity groupings. In addition, statistical subgroup inferential analyses could be conducted in further research with larger samples as this was beyond the scope of this research study. In addition, this response rate is within a 6% margin of error at the 95% level of confidence, which meets recommended levels [23]. More specifically, Nulty [23] states that “response rates to online surveys of teaching and courses are nearly always very much lower than those obtained when using on-paper surveys”. Lastly, the study employed a self-report questionnaire survey, which was perceived as pertinent given the sensitive nature of the area under study, although there may be instances of social desirability bias. It is recommended that a social desirability questionnaire be used in conjunction with the GWHQ given that the patriarchal and colonial history of medicine may create high levels of conformity, and students may be unaware of the unconscious influences associated with these levels of conformity [24]. In addition, we propose qualitative research will likely be useful to explore and contextualize some of the areas that are being reported.

## 5. Conclusions

The findings of this study indicate that the 24-item GWHQ, reduced from the original validated 25-item set, is a useful and valid instrument for appraising instances of harassment within clinical placements attended by medical students in New Zealand. The four latent domains, namely Verbal Hostility, Covert Hostility, Manipulation, and Physical Hostility, can be used to authentically describe levels of workplace toxicity and unprofessional behavior within a clinical learning environment. This will also enable interventions to be implemented and thoroughly assessed and this will likely further prevent the sequela oftentimes associated with toxic interactions, such as psychological and physical health concerns. Therefore, the GWHQ has utility in terms of informing human resource personnel in healthcare settings, clinical educators, student advocates, and future research in this area.

## Figures and Tables

**Table 1 behavsci-13-00791-t001:** CFA factor loadings.

Item	Factor Loadings			
	Verbal Hostility	Covert Hostility	Manipulation	Physical Hostility
1. yelled	0.60			
2. gossiped, rumors	0.70			
3. comments, intelligence	0.71			
4. pressured	0.73			
5. gestures	0.73			
6. “troublemaker”	0.68			
7. humiliated	0.69			
11. swore at you	0.49			
13. comments, personality	0.75			
14. talked down to	0.81			
15. treated less good	0.78			
16. blamed you	0.70			
8. took credit		0.58		
9. ignored you		0.56		
10. interrupted you		0.79		
23. treated unfairly		0.72		
26. work wasn’t part		0.71		
27. excluded you		0.70		
12. turned against			0.84	
22. left notes			0.87	
28. threatened			0.79	
18. pushed or grabbed				0.79
25. threw something				0.91
29. hit physically				0.96

Notes: (1) the original item numbering has been retained, (2) item 19 has been removed from the 25-item list, (3) item descriptors have been condensed, for full details see the original article, [6].

**Table 2 behavsci-13-00791-t002:** Correlations between the four subscales of the GWHQ. All coefficients are Spearman’s rho and were significant at *p* < 0.01.

	Covert Hostility	Manipulation	Physical Hostility
Verbal Hostility	0.75	0.48	0.18
Covert Hostility	-	0.39	0.22
Manipulation		-	0.22

## Data Availability

The data used to support the findings of this study are available from the corresponding author upon request.

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
