# Peer review of "Validation of the Generalized Workplace Harassment Questionnaire for Use with Medical Students"

_behavsci, 2023, doi:10.3390/bs13100791_

Round 1

Reviewer 1 Report

Excellent work but I miss gender view. According to the concept of politics as decribed by Millett Kate "The term "politics" shall refer to power-structured relationships, arangements whereby one group of persons is controlled by another (p. 23, 2016) and how this is a definition or agression and violence os some men groups towars anothers, The power concep is a key to be included. I would also recomended to include phobias in a explicit way as they are dagerous to mental health and gender sexism. Invisibility of handicaped boys and girls. I agree with the metodology.

Author Response

The primary purpose of this study was to conduct a psychometric evaluation of the GWHQ using confirmatory factor analysis techniques, hence focussing on statistical evaluations of domain structure and item fit. Nonetheless, the issue raised by this review is interesting and we have added a comment in Discussion, i.e., “Data obtained from the GWHQ can be combined with other measures to create a more detailed picture of organizational workplace dynamics, thus identifying whether marginalized groups are more at risk of harassment, and whether harassment can have consequential outcomes such as toxic team-based behaviors, power imbalance, psychological and physical health issues, lowered performance and safety concerns.” This sentence acknowledges the importance of power in this area of research. Once this questionnaire has been validated it can be used with more confidence in medical education to explore areas of power imbalance, and this would apply to future subgroup inferential analyses.

Reviewer 2 Report

Overall, the manuscript addresses a relevant issue; it is clearly written

Author Response

Thank you for your appreciation of the article.

Reviewer 3 Report

This manuscript presents the results of a research work which used the Generalized Workplace Harassment Questionnaire (GWHQ) to assess experiences of harassment or hostility in clinical placements attended by medical students in New Zealand. The results of a confirmatory factor analysis showed that a 28-item GWHQ with four factors (verbal hostility, covert hostility, manipulation, and physical aggression) had good fit. The authors conclude that the GWHQ is a valid instrument for appraising instances of harassment or hostility in clinical placements for medical students in New Zealand.

The manuscript is well-organized and to the point. However, I have some comments and suggestions for improving this work.

1.Introduction

The authors could explain how the GWHQ was developed and even modified to be used in different workplaces including medical schools [refs 3-5].This would help to justify their aim to validate this research instrument and make it more relevant to the wider literature. For example, they could cite the work of Lin, T. W., Rospenda, K. M., & Richman, J. A. (2022).Relationships between school harassment and problematic drinking in a college sample: Is need for approval a moderator?. Journal of American College Health, 1-9).

2. Methodology

Lines 111-117:  If the difference in the proportion of male and female students is statistically significant, then it is possible that the gender ratio does vary between year of study or ethnic group. However, the author should also consider whether the difference is large enough to be meaningful. Perhaps a X2 test would resolve this issue.

3Discussion

Overall, the discussion is a good contribution to the literature on harassment in clinical settings. However, the discussion could be improved by providing more specific evidence to support its claims and by discussing the potential impact of the limitations of the study. More specifically,

3.1.The discussion could be more specific about the implications of the findings for clinical leaders, curriculum planners, and other stakeholders. For example, the discussion could discuss how the findings could be used to identify and address toxic learning environments.

3.2.The authors state that the GWHQ is a reliable and valid instrument for measuring harassment in clinical settings for medical students. The authors could better demonstrate that in agreement with previous research in other settings, the GWHQ is a reliable and valid research tool  is a reliable and valid instrument for measuring harassment in clinical settings for medical students

3.3.The authors recommend that a social desirability questionnaire be used in conjunction with the GWHQ. However, the discussion does not explain why a social desirability questionnaire is necessary. It would be helpful if the authors  explained how a social desirability questionnaire could help to reduce bias in the results.

Author Response

The primary purpose of this study was to conduct a psychometric evaluation of the GWHQ using confirmatory factor analysis techniques, hence focussing on statistical evaluations of domain structure and item fit. Nonetheless, we have added more detail regarding the current usage and value of the GWHQ (in Introduction) and included the Lin et al article cited by this reviewer. We feel this has given the article more breadth beyond our statistical focus.

We did not compute any inferential subgroup analyses given the focus of the paper was on validation. However given this reviewers’ comments we have made several references in the paper suggesting that once the questionnaire is validated, subgroup inferential analyses could be conducted. We aimed to focus on the samples as a whole to determine model fit of the questionnaire. The useful suggestion by this reviewer of more intricate subgroup analyses could be applied to larger samples once the questionnaire has been validated for this cohort.   

In Discussion, we have added more detail on the implications of the results, without depleting our statistical focus of the study. For us, the light shone mainly on whether the questionnaire could be validated. Nonetheless, we have now added more detail to show how it could be useful once it has been validated. For example, “Data obtained from the GWHQ can be combined with other measures to create a more detailed picture of organizational workplace dynamics, thus identifying whether marginalized groups are more at risk of harassment, and whether harassment can have consequential outcomes such as toxic team-based behaviors, power imbalance, psychological and physical health issues, lowered performance and safety concerns.”

Reviewer 4 Report

The authors have a solid research career.The authors need to improve:

1. Justify medical students receiving more pressure or harassment in their training environment. 

2. Determine whether it is workplace bullying or clinical practice training, university context.

3.  Reinforce the introduction with the analysis of previous and related studies on the Generalised Workplace Bullying Questionnaire. 

3. Expand references 

4. Expand key words 

Author Response

The primary purpose of this study was to conduct a psychometric evaluation of the GWHQ using confirmatory factor analysis techniques, hence focussing on statistical evaluations of domain structure and item fit. Nonetheless, we have added more detail regarding the current usage and value of the GWHQ (in Introduction). We have included an explicit link to our previous scoping review in this area that details the antecedents (e.g., being exposed to toxic workplace cultural practices) and consequences (e.g., depleted physical and psychological health) of harassment with specific reference to the medical students’ clinical experience.  

For us, the light shone mainly on whether the questionnaire could be statistically validated. Nonetheless, we have now added more explanatory notes on how the GWHQ could be applied and added more current literature using this instrument to ensure the narrative has more depth.  For example, “Data obtained from the GWHQ can be combined with other measures to create a more detailed picture of organizational workplace dynamics, thus identifying whether marginalized groups are more at risk of harassment, and whether harassment can have consequential outcomes such as toxic team-based behaviors, power imbalance, psychological and physical health issues, lowered performance and safety concerns.”